# Chemical Diversity of Mo_5_S_5_ Clusters with Pyrazole: Synthesis, Redox and UV-vis-NIR Absorption Properties

**DOI:** 10.3390/ijms241813879

**Published:** 2023-09-09

**Authors:** Iulia V. Savina, Anton A. Ivanov, Ilia V. Eltsov, Vadim V. Yanshole, Natalia V. Kuratieva, Andrey Y. Komarovskikh, Mikhail M. Syrokvashin, Michael A. Shestopalov

**Affiliations:** 1Nikolaev Institute of Inorganic Chemistry, Siberian Branch of Russian Academy of Sciences, 3 Acad. Lavrentiev Ave., Novosibirsk 630090, Russia; savina@niic.nsc.ru (I.V.S.); ivanov338@niic.nsc.ru (A.A.I.); kuratieva@gmail.com (N.V.K.); komarovskikh@niic.nsc.ru (A.Y.K.); syrokvashin@niic.nsc.ru (M.M.S.); 2Department of Natural Sciences, Novosibirsk State University, 1 Pirogova St., Novosibirsk 630090, Russia; eiv@fen.nsu.ru; 3Department of Physics, Novosibirsk State University, 1 Pirogova St., Novosibirsk 630090, Russia; vadim.yanshole@tomo.nsc.ru; 4International Tomography Center SB RAS, 3a Institutskaya Str., Novosibirsk 630090, Russia

**Keywords:** molybdenum, metal cluster, pyrazole, EPR, NMR, cyclic voltammetry, UV-vis-IR absorption

## Abstract

The chemistry of transition metal clusters has been intensively developed in the last decades, leading to the preparation of a number of compounds with promising and practically useful properties. In this context, the present work demonstrates the preparation and study of the reactivity, i.e., the possibility of varying the ligand environment, of new square pyramidal molybdenum chalcogenide clusters [{Mo_5_(μ_3_-S)^i^_4_(μ_4_-S)^i^(μ-pz)^i^_4_}(pzH)^t^_5_]^1+/2+^ (pzH = pyrazole, i = inner, t = terminal). The one-step synthesis starting from the octahedral Mo_6_Br_12_ cluster as well as the substitution of the apical pyrazole ligand or the selective bromination of the inner pyrazolate ligands were demonstrated. All the obtained compounds were characterized in detail using a series of physicochemical methods both in solid state (X-ray diffraction analysis, etc.) and in solution (nuclear magnetic resonance spectroscopy, mass spectrometry, etc.). In this work, redox properties and absorption in the ultraviolet-visible and near-infrared region of the obtained compounds were studied.

## 1. Introduction

Investigation of functional compounds and materials is always related to the problems of modern mankind, such as solving energy problems, creating effective drugs, improving industrial processes, and so on. In this context, transition metal cluster complexes [1] are of great interest due to their structural diversity [2,3,4,5], chemical modification [4], and combination of useful properties [6,7,8,9]. Among such clusters, molybdenum compounds can be particularly distinguished, whose representatives have proven to be useful, for example, as agents in biology and medicine [6], components of optical materials [9,10], catalysts [11], etc. The most studied in these fields are the octahedral halide clusters [Mo_6 × 8_L_6_] (X = Cl, Br, I; L = organic or inorganic ligands) [12] and the tri- and tetranuclear chalcogenide clusters with cluster cores {Mo_3_Q_7_}, {Mo_3_Q_4_} and {Mo_3_Q_4_M′} (Q = S, Se; M′ = Cu, Ni, Co, Fe, etc.) [4,11]. For example, materials doped with an octahedral halide molybdenum clusters show high antimicrobial activity against five common pathogens (bacteria and fungi) under white light irradiation [13] or can be used as nontoxic photoluminescent agents for bioimaging and cell delivery [14]. On the other hand, nanoaggregates of molecular clusters are known to possess blue light phototoxic effect against HeLa cells at nanomolar concentrations and antibacterial effect against several pathogenic strains [15]. Cubane-type Mo_3_NiS_4_ clusters showed high catalytic activity in the intermolecular cyclization of various alkynoic acids [16], while {Mo_3_S_4_} clusters combined with polyoxometalate were found to be highly effective in the hydrogen evolution reaction at 0 V vs. reversible hydrogen electrode [17]. In another study, cuboidal trinuclear and tetranuclear metal−chalcogenido clusters were shown to be efficient optical limiters [18]. Studies of molybdenum cluster compounds are not limited to the above complexes and applications and concern clusters of different nuclearity and structure [4,19,20,21,22,23]. However, there is much less work demonstrating their properties and potential applications. Nevertheless, the development of new cluster compounds and the study of their properties will indeed lead to new promising functional materials.

In this context, we have recently demonstrated the preparation of a new representative of chalcogenide molybdenum clusters, namely square pyramidal [{Mo_5_(μ_3_-Se)^i^_4_(μ_4_-Se)^i^(μ-pz)^i^_4_}(pzH)^t^_5_]^1+/2+^ (i = inner, t = terminal (basal and apical), denoted as **Mo_5_Se_5_^red/ox^**) [24]. Clusters obtained in two steps (via an amorphous intermediate of unknown structure “NaMo_6_Se_8_Br_4_”) from Mo_6_Br_12_ in low yield (about 30%) show reversible redox transitions between paramagnetic and diamagnetic states. The aim of this work is to extend the family of square pyramidal clusters to demonstrate the chemical modification and to study the physicochemical properties. Novel S-containing clusters [{Mo_5_(μ_3_-S)^i^_4_(μ_4_-S)^i^(μ-pz)^i^_4_}(pzH)^t^_5_]^1+/2+^ (denoted as **[1^red/ox^]**) were obtained in one step with high yield (~60%) starting from Mo_6_Br_12_ (Figure 1). The ligand environment can be modified in two ways: (i) interaction with HBr resulting in selective substitution of the apical pyrazole ligand by Br-ligand to form [{Mo_5_(μ_3_-S)^i^_4_(μ_4_-S)^i^(μ-pz)^i^_4_}(pzH)^bs^_4_Br^a^]^+^ (denoted as [2], bs = basal, a = apical) or (ii) interaction with Br_2_ resulting in selective bromination of the bridging pyrazolate ligands to form [{Mo_5_(μ_3_-S)^i^_4_(μ_4_-S)^i^(μ-4-Br-pz)^i^_4_}(pzH)^t^_5_]^2+^ (denoted as [3]) (Figure 1). It is also possible to perform a sequential modification of clusters, combining both bromination of pyrazolate ligands and substitution of the apical ligand, resulting in the compound [{Mo_5_(μ_3_-S)^i^_4_(μ_4_-S)^i^(μ-4-Br-pz)^i^_4_}(pzH)^bs^_4_Br^a^]^+^ (denoted as [4]). All the compounds obtained were characterized in detail using a series of physicochemical methods of analysis, both in the solid state (single-crystal and powder X-ray diffraction analysis, elemental analysis, FTIR, etc.) and in solution (NMR spectroscopy and mass spectrometry). The influence of the modification of the cluster complexes on the redox properties as well as on the absorption in the ultraviolet-visible (UV-vis) and near-infrared (NIR) region is demonstrated.

## 2. Results and Discussion

### 2.1. Synthesis and General Characterization of Square Pyramidal Clusters

As previously reported, selenide square pyramidal molybdenum clusters can be obtained from the bromide octahedral cluster Mo_6_Br_12_ in two steps: (i) substitution of halogens in the cluster core by interaction with NaHSe in water to form a black amorphous insoluble compound of unknown structure; (ii) interaction of the obtained product with the pyrazole melt in a sealed ampoule at 200 °C [24]. Since the synthesis proceeds in two steps through a product of unknown structure, it is rather difficult to guess which step the cluster core is oxidized at (Mo^II^_6_ → Mo^III^_5_), and a vertex of the octahedron is removed. The replacement of NaHSe by NaHS in the first step also leads to the formation of a black amorphous product, but when it is introduced into the reaction in the pyrazole melt, no formation of the necessary products is observed. Therefore, in this work, we proposed a method that combines the two above steps, namely the one-pot substitution of all ligands of a halide cluster with the formation of a five-nuclear cluster—the interaction of Mo_6_Br_12_ with a chalcogen source and pyrazole in a sealed ampoule.

When choosing the conditions, the following points were considered: (i) during the reaction, molybdenum is oxidized, i.e., an oxidant is needed; (ii) elimination of one molybdenum to form an unknown compound; (iii) the source of sulfur in cluster core is S^2−^; (iv) pyrazole is both the reaction medium and the organic ligand. An analysis of the literature indicates the participation of H^+^ [25] or polychalcogenides Q_x_^2−^ [26] as the main oxidants in similar reactions of molybdenum chalcogenide cluster formation in solutions. Therefore, in order to find the optimal reaction conditions, the syntheses were carried out at 200 °C for 2 days in a sealed ampoule between Mo_6_Br_12_ (200 mg) and pyrazole in a molar ratio of 1:20 in the presence of different chalcogen sources: Na_2_S, NaHS (varying molar ratio from 5 to 10 for 1 cluster) and their combination with S (varying the molar ratio from 2 to 5 for 1 cluster). In almost all cases, the formation of the required pentanuclear product was observed, but the yields were often not more than 10%. The best result was obtained in the molar ratio Mo_6_Br_12_:Na_2_S:S:pzH = 1:6:3.5:20 in the synthesis at 200 °C for 2 days in a sealed ampoule. It is also important to note the use of dried Na_2_S in the synthesis, since a large amount of water in the reaction mixture leads to the destruction of the reaction product. Slow cooling of the reaction mixture results in the formation of dark green crystals [{Mo_5_(μ_3_-S)^i^_4_(μ_4_-S)^i^(μ-pz)^i^_4_}(pzH)^t^_5_]Br·pzH·H_2_O (denoted as **[1^red^]Br**) suitable for single-crystal X-ray diffraction analysis (SCXRD), which were washed with diethyl ether and manually separated from the byproducts. The easiest way to completely isolate the desired product is to dissolve it in an organic solvent. However, during dissolution, for example in acetonitrile, the cluster core is oxidized to form the cationic complex [{Mo_5_(μ_3_-S)^i^_4_(μ_4_-S)^i^(μ-pz)^i^_4_}(pzH)^t^_5_]^2+^. By dissolving the product in acetonitrile, evaporating the solution, and dissolving in dichloromethane, any byproducts can be separated to isolate the compound [{Mo_5_(μ_3_-S)^i^_4_(μ_4_-S)^i^(μ-pz)^i^_4_}(pzH)^t^_5_]Br_2_ 2H_2_O (denoted as **[1^ox^]Br_2_**) with a yield of about 60%. According to the energy-dispersive X-ray spectroscopy (EDS) data, the remaining black amorphous byproduct, which is insoluble in acetonitrile, contains molybdenum and sulfur in a ratio of 1:2, which can be attributed to the formation of MoS_2_, presumably upon elimination of a Mo vertex of the octahedron Mo_6_. Based on all the results, the formation of a new cluster can be described by the following reaction:2Mo^II^_6_Br_12_ + 12Na_2_S + 7S + 18pzH → 2[Mo^III^_5_S_5_(μ-pz)_4_(pzH)_5_]Br + 2Mo^IV^S_2_ + 22NaBr + 2NaHS + 3H_2_S

Note that in this reaction, elemental sulfur (or Na_2_S_n_, which can be formed under such conditions) acts as an oxidizing agent for the transition of Mo^II^ to Mo^III^. To confirm the importance of sulfur in this reaction, an additional experiment was performed in which elemental sulfur was replaced by elemental selenium. Despite the significant difference in the redox properties of the elemental compounds, a green product was isolated in this reaction (yield~90 mg starting from 200 mg of Mo_6_Br_12_), which, according to high-resolution electrospray mass spectrometry (HR-ESI-MS), is a mixture of compounds with the composition [{Mo_5_S_5−x_Se_x_(μ-pz)_4_}(pzH)_5_]^2+^ (x = 0, 1, 2, 3) (Figure 2). Thus, in addition to the role of oxidant, elemental sulfur/selenium also acts as source of inner ligands.

As noted above, upon dissolution, the **[1^red^]Br** complex transforms to **[1^ox^]Br_2_**, i.e., the 15 VEC (valence electron per cluster) paramagnetic complex becomes 14 VEC diamagnetic one (to be discussed further) by analogy with the published selenium counterpart [24]. The diamagnetic nature of **[1^ox^]Br_2_** allows for its characterization in solution using NMR (Figure 3a and Appendix A). The ^1^H NMR spectrum of the compound contains a group of eight signals corresponding to all the protons of the pyrazole and pyrazolate ligands. The positions of the signals and their ratios are consistent and very close to those for a similar selenium cluster. In short, when coordinated, the proton signals of the pyrazolate ligand are slightly shifted downfield (Δ~0.1 ppm) relative to free pyrazole, and the basal pyrazole proton signals are strongly shifted downfield (Δ~0.6 and 0.7 ppm for H4 and H5 correspondingly), with the largest shift in the H3 proton signal (Δ~1.2 ppm), which is closest to the metal center, further confirming the coordination. Only the apical pyrazole signals are shifted upfield (Δ~0.4 ppm for H7 and H8), with the largest shift in the H6 proton signal (Δ~1.3 ppm) being closest to the coordination. A different shift in the signals of the terminal (basal and apical) pyrazoles indicates different interactions with the cluster and may contribute to the different reactivity of the ligands. The composition of the cluster was also confirmed using HR-ESI-MS (Figure 3b and Appendix A): the spectra contain the forms {[{Mo_5_S_5_(μ-pz)_4_}(pzH)_5_]Br}^+^ (m/z = 1328.6111) and {[{Mo_5_S_5_(μ-pz)_4_}(pzH)_5_]}^2+^ (m/z = 624.8470), as well as less intense forms with solvent molecules or with a smaller number of terminal ligands, which are apparently removed during the ionization process.

The ligand environment of the **[1^ox^]Br_2_** complex can be modified in two ways (Figure 4): (i) selective replacement of the apical pyrazole ligand with a bromine anion to form [{Mo_5_(μ_3_-S)^i^_4_(μ_4_-S)^i^(μ-pz)^i^_4_}(pzH)^bs^_4_Br^a^]Br (denoted as **[2]Br**); (ii) selective brominating of pyrazolate ligands at the fourth position to form [{Mo_5_(μ_3_-S)^i^_4_(μ_4_-S)^i^(μ-4-Br-pz)^i^_4_}(pzH)^t^_5_]Br_2_ (denoted as **[3]Br_2_**).

The first approach was realized by the interaction of **[1^ox^]Br_2_** with concentrated hydrobromic acid in methanol. Note that this process can be carried out in the presence of other acids (e.g., acetic acid, phosphoric acid), including prolonged soaking in methanol, which is itself a weak acid. Previously, when studying the [{Mo_5_(μ_3_-Se)^i^_4_(μ_4_-Se)^i^(μ-pz)^i^_4_}(pzH)^t^_5_]^1+/2+^ clusters, we also observed the substitution of the apical pyrazole upon standing in methanol, which we suggested was related to the coordination of methanol [24]. Apparently, in this case, the apical pyrazole is more labile under “acidic” conditions and the coordination of the Br anion, which is always present as a counterion in Mo_5_ complexes, is most preferred. This equilibrium can be completely shifted to the complex with the apical bromine ligand by the addition of hydrobromic acid. The second modification approach was realized by the interaction of **[1^ox^]Br_2_** with Br_2_ in methylene chloride. In this reaction, the hydrogen atom in the fourth position of the pyrazolate ligand is electrophilically substituted with bromine. In both cases, other ligands are not involved in the reactions.

Both modifications were confirmed using complex physicochemical analysis (see Section 3) and in solution using NMR spectroscopy (Figure 5 and Appendix A) and mass spectrometry (Appendix A). Thus, in the ^1^H NMR spectrum of **[2]Br**, there are no signals from the apical pyrazole, while the signals of the remaining ligands are preserved and slightly upfield shifted (Figure 5). Upon bromination of the bridging pyrazolate ligands, the H2 proton signal disappears completely, the H1 proton signal is shifted downfield (Δ~0.2 ppm) and its multiplicity changes from a doublet to a singlet (Figure 5), and the C2 carbon signal (the carbon in the fourth position of the pyrazolate ligand) is strongly shifted upfield (Δ~10 ppm), which is explained by the effect of a heavy nuclei, manifested by the shielding of the nearest carbon nuclei by the bromine electron shell. All these observations support the selective bromination of the pyrazole at the 4-position. HR-ESI-MS of solutions of the compounds **[2]Br** and **[3]Br_2_** contain the forms {[{Mo_5_S_5_(μ-pz)_4_}(pzH)_4_Br]}^+^ (m/z = 1260.5738) and {[{Mo_5_S_5_(μ-4-Br-pz)_4_}(pzH)_5_]Br}^+^ (1576.1998), respectively (Appendix A), further confirming the composition of the cluster complexes.

The final stage of modification of the ligand environment of the five-nuclear cluster was carried out by successively implementing the above approaches: bromination of the pyrazolate ligands and replacement of the apical pyrazole with bromine, yielding [{Mo_5_(μ_3_-S)^i^_4_(μ_4_-S)^i^(μ-4-Br-pz)^i^_4_}(pzH)^bs^_4_Br^a^]Br (denoted as **[4]Br**). Performing a sequential modification is no different from performing each modification separately. The reaction product was also characterized in detail using a number of analytical methods (see Materials and Methods), including studies in solution. The absence of apical ligand signals in the ^1^H NMR spectrum confirms the substitution of the ligand with bromine, and the preservation of the signals of the remaining ligands indicates the absence of effect of this modification on functionalized pyrazolate ligands (Figure 6 and Appendix A). The mass spectrum of the compound also contains the necessary forms, for example, {[{Mo_5_S_5_(μ-4-Br-pz)_4_}(pzH)_4_Br]}^+^ (Figure 6 and Appendix A).

### 2.2. Crystal Structure of Compounds

The crystal structure of all the compounds obtained was determined using single-crystal X-ray diffraction analysis (SCXRD).
[{Mo_5_(μ_3_-S)^i^_4_(μ_4_-S)^i^(μ-pz)^i^_4_}(pzH)^t^_5_]Br pzH H_2_O 

The crystals of **[1^red^]Br** suitable for SCXRD were selected from the reaction mixture. The compound crystallizes in the orthorhombic space group *P*ccn and contains the cationic cluster complex [{Mo_5_(μ_3_-S)^i^_4_(μ_4_-S)^i^(μ-pz)^i^_4_}(pzH)^t^_5_]^+^, bromine anion, and co-crystallized water and pyrazole molecules. The cluster complex is a square Mo_5_ pyramid with four triangular-faced μ_3_-S and one base-faced μ_4_-S. In addition, four bridging pyrazolate ligands coordinated to the base of the pyramid act as inner ligands, forming a cluster core {{Mo_5_(μ_3_-S)^i^_4_(μ_4_-S)^i^(μ-pz)^i^_4_}^+^. The molybdenum coordination sphere is completed by terminal pyrazole ligands (four basal (bs) and one apical (a)) (Figure 7). In this organization, the inner pyrazole ligands are at an angle of ~45 degrees to the base of the pyramid (Figure 7), and the terminal basal ligands are nearly parallel to the base of the pyramid. The apical pyrazole ligand is disordered over four positions (0.067, 0.067, 0.433 and 0.433) due to free rotation around the Mo-N bond (Figure 7).

The bromine anion and the water molecule have an occupancy of 0.5 and occupy a close position. The hydrogen atom of the N-H group of the solvate molecule also has an occupancy of 0.5. Basal ligands of cluster cations interact with a bromine anion or a water molecule through an N-H⋯Br/O hydrogen bond (N⋯Br/O distances of 2.91–3.42 Å). Such fragments are linked through the solvate pyrazole molecule by Br/O⋯H-N or O-H⋯N hydrogen bonds (Br/O⋯N distances of 2.93–3.32 Å), forming infinite chains along the *a*-axis (Appendix A). The chains are connected by π–π stacking interactions between basal ligands of adjacent clusters (distances between centers of 3.54 Å), forming layers along the *ab* plane (Appendix A). The layers are packed into a three-dimensional structure by C-H⋯π interactions between the C-H groups of the apical ligand and the π system of the pyrazolate ligands of the adjacent cluster (distances between center and H of 2.82–2.91 Å) (Appendix A).
[{Mo_5_(μ_3_-S)^i^_4_(μ_4_-S)^i^(μ-pz)^i^_4_}(pzH)^t^_5_][Mo_6_I_14_]·4CH_3_CN 

We did not succeed in obtaining single crystals of the compound **[1^ox^]Br_2_** suitable for SCXRD. However, the cationic complex was crystallized with the cluster anion [{Mo_6_I_8_}I_6_]^2−^ to form [{Mo_5_(μ_3_-S)^i^_4_(μ_4_-S)^i^(μ-pz)^i^_4_}(pzH)^t^_5_][Mo_6_I_14_]·4CH_3_CN (denoted as **[1^ox^][Mo_6_I_14_]**). The compound crystallizes in the monoclinic space group *C* 2/c. When the cluster is oxidized, the Mo_5_ square pyramid is slightly distorted (to be discussed later), while the organization of the cationic cluster complex is preserved. The apical pyrazole ligand is disordered in two positions (0.5 and 0.5) that overlap each other and are realized by free rotation of the ligand by 180 around the Mo-N bond (Appendix A). Cluster cations and solvate molecules of acetonitrile pack into layers along the *ab* plane (Appendix A). Another layer of cluster anions [{Mo_6_I_8_}I_6_]^2−^ and solvate molecules of acetonitrile are located between the layers (Appendix A). Basal ligands of cluster complexes form N-H⋯N hydrogen bonds with acetonitrile molecules of the adjacent layer (N⋯N distances of 2.94–3.31 Å). There are no other significant interactions between the layers.
[{Mo_5_(μ_3_-S)^i^_4_(μ_4_-S)^i^(μ-pz)^i^_4_}(pzH)^bs^_4_Br^a^]Br·2.5H_2_O·0.5DMF 

Single crystals of the compound **[2]Br** were obtained by diffusion of diethyl ether into a solution of the complex in a mixture of DMF and methanol (90:10 vol.%). The compound [{Mo_5_(μ_3_-S)^i^_4_(μ_4_-S)^i^(μ-pz)^i^_4_}(pzH)^bs^_4_Br^a^]Br·2.5H_2_O·0.5DMF (denoted as **[2]Br**·2.5H_2_O·0.5DMF) crystallizes in the monoclinic space group *C* 2. The structure of the cluster cation [2]^+^ differs from **[1^ox^]**^2+^ by replacing of the apical pyrazole ligand with bromine (Figure 8). The structure contains two bromine anions with position occupancy of 0.5 each, beside which there are water molecules also with position occupancy of 0.5 (similar to the **[1^red^]Br** structure). The cluster cations form two types of chains (along axes *a* and *c*) in which the complexes are connected by hydrogen bonds N-H⋯O/Br (N⋯O/Br distances of 2.89–2.96 Å/3.28–3.47 Å), C-H⋯Br (C⋯Br distances of 3.35 Å) and O-H⋯Br (O⋯Br distances of 3.38–3.42 Å) between the basal pyrazole ligands, bromine anions and solvate water molecules (Appendix A). The chains are linked by π–π stacking interactions between the basal pyrazole ligands involved in hydrogen bonds (distances between centers of 3.67–3.79 Å, Appendix A) or pyrazolate ligands of adjacent clusters (distances between centers of 3.45 Å, Appendix A). The solvate molecules of N,N-dimethylformamide (position occupancy 0.5) and water (position occupancies 0.25 and 0.5) are located in the free space between the chains.
[{Mo_5_(μ_3_-S)^i^_4_(μ_4_-S)^i^(μ-4-Br-pz)^i^_4_}(pzH)^t^]Br_2_·CH_3_CN·Et_2_O 

Single crystals of the compound **[3]Br_2_** were obtained by diffusion of diethyl ether into a solution of the complex in acetonitrile. The compound [{Mo_5_(μ_3_-S)^i^_4_(μ_4_-S)^i^(μ-4-Br-pz)^i^_4_}(pzH)^t^]Br_2_·CH_3_CN·Et_2_O (denoted as **[3]Br_2_**·CH_3_CN·Et_2_O) crystallizes in the monoclinic space group *P* 2_1_/n. The structure of the cluster cation [3]^2+^ differs from **[1^ox^]**^2+^ by the presence of bromine substituents in the fourth position of the pyrazolate ligands (Figure 9). The apical pyrazolate ligand is disordered in two positions (0.5 and 0.5) that overlap each other and are realized by free rotation of the ligand by 180 degrees around the Mo-N bond (Figure 9). In the structure, the cluster cations are linked by hydrogen bonds between terminal pyrazole ligands and bromine anions: two adjacent basal pyrazole ligands form N-H⋯Br bonds (N⋯Br distances of 3.23 Å) with one bromine and C-H⋯Br bonds (C⋯Br distances of 3.82 Å) with another bromine, forming infinite chains along the *b*-axis (Appendix A). Interchain bonding is realized by apical pyrazole ligands that form C-H/N-H⋯Br (C/N⋯Br distances of 3.41 Å) hydrogen bonds with neighboring chains, forming layers along the *ab* plane (Appendix A). Thus, each bromine anion is associated with three clusters (two from the same chain and one from the neighboring chain). Due to this organization, the layers are decorated with pyrazolate ligand bromines (Appendix A). No significant interactions between the layers were observed. Solvate molecules of acetonitrile are located in free space in the cluster layers, and diethyl ether is located between the layers.
[{Mo_5_(μ_3_-S)^i^_4_(μ_4_-S)^i^(μ-4-Br-pz)^i^_4_}(pzH)^bs^_4_Br^a^]Br·1.25H_2_O·2DMSO 

Single crystals of the compound **[4]Br** were obtained by diffusion of ethyl acetate into a solution of the complex in dimethyl sulfoxide. The compound [{Mo_5_(μ_3_-S)^i^_4_(μ_4_-S)^i^(μ-4-Br-pz)^i^_4_}(pzH)^bs^_4_Br^a^]Br·1.25H_2_O·2DMSO (denoted as **[4]Br**·1.25H_2_O·2DMSO) crystallizes in the monoclinic space group *P* 2_1_/m. The structure of the cluster cation [4]^+^ differs from **[1^ox^]**^2+^ in the presence of bromine substituents in the fourth position of the pyrazolate ligands and a bromine ligand instead of a pyrazole at the apical position (Figure 10). The structure contains two bromine anions, each disordered over three positions (site occupancies 0.125, 0.0625 and 0.0625), resulting in a total of one bromine anion per cluster cation. The cluster cations are linked in chains along *a*-axis via C-H⋯Br hydrogen bonds (C⋯Br distances of 3.23–3.40 Å) between the pyrazolate ligand of one cluster, the bromine anion and the basal pyrazole of the other cluster (Appendix A). The chains are linked by weak C-H⋯Br-C hydrogen bonds (C⋯Br distances of 3.87 Å) between the pyrazolate ligands of adjacent clusters (Appendix A). Basal pyrazoles form N-H⋯O hydrogen bonds (N⋯O distances of 2.86–2.91 Å) with solvate DMSO molecules and C-H⋯Br (C⋯Br distances of 3.21 Å) with other bromine anions. Solvate water molecules with partial position occupancy are located in the free space between the chains.

The resulting series of crystal structures allows us to study the effect of modifications on the structure of the five-core clusters. The main bond lengths in the cluster cations are summarized in Table 1. Comparing the structure of the new S-containing clusters with the previously published selenide complexes, one can observe a decrease in the size of the Mo_5_ pyramid in all directions, which is typical for other cluster complexes depending on the chalcogen in the cluster core [27]. The oxidation of the cluster core (**[1^red^]**^+^ to **[1^ox^]**^2+^) leads to a slight broadening of the pyramid base and a decrease of the distances in the lateral faces, i.e., a flattening of the pyramid. Bromination of pyrazolate ligands (**[1^ox^]**^2+^ to [3]^2+^) has virtually no effect on the distances in the cluster complex, since the modification affects groups far from the metal core. The substitution of the apical pyrazole with bromine has different effects: (i) in the transition from **[1^ox^]**^2+^ to [2]^+^, there is a slight decrease in the distances at the base of the pyramid, while all other bond lengths remain the same; (ii) in the transition from [3]^2+^ to [4]^+^, a more significant change in the distances in the Mo_5_ pyramid (they become close to **[1^red^]**^+^) and an elongation of the Mo-N^μ-pz^ bonds are observed. However, significant distortions in the structures with different modifications are not observed.

### 2.3. Physicochemical Properties

According to the charge balance, in the compound **[1^red^]Br**, all molybdenum has charge state +3, i.e., the pyramid Mo_5_ contains 15 valence electrons per cluster (VEC). In this case, the clusters have one unpaired electron, as has been demonstrated for the Se-containing cluster [24]. Thus, in this work, the magnetic properties of **[1^red^]Br** were studied using magnetic susceptibility measurements and electron paramagnetic resonance (EPR) spectroscopy. The measured magnetic susceptibility temperature dependence χ(T) is shown in Figure 11. The linear behavior of the inverse magnetic susceptibility 1/χ(T) (inset in Figure 11) allows one to conclude that the χ(T) obeys the Curie–Weiss law χ(T) = C/(T-Θ)^−1^ within the temperature range of 80–300 K (green dashed line in Figure 11). The negative sign of the Weis constant Θ = –48(5)K indicates antiferromagnetic ordering at low temperatures. The calculated effective moment value of μ_eff_ = 2.19(3) μ_B_ corresponds to a spin of s_g_ = 0.67(1). Using the isotropic value of the g-tensor (g_iso_ = 2.07) obtained using the EPR method (see below), the number of unpaired electrons per cluster was determined to be 1.34(2), which is close to theoretical value (one unpaired electron).

In the X-frequency band, the EPR spectra of the **[1^red^]Br** polycrystalline sample have an appearance corresponding to the species with spin S = ½. While at room temperature the EPR lines are broadened (Appendix A), at 77 K the spectrum reveals the rhombic symmetry of the g-tensor (Figure 11). The experimental data have been fitted with a Lorentzian line shape and a conventional spin Hamiltonian using the least squares method:$$\hat{H}=\beta \hat{S}gH$$
where β is a Bohr magneton, g is a g-tensor, and H is a magnetic field. At 77 K, the principal values of the g-tensor have been obtained equal to g_xx_ = 2.12, g_yy_ = 2.11, and g_zz_ = 1.99 (g_iso_ = 2.07). The estimation from the weight of the sample portion and the molar mass provides 1.1 spins per cluster, quite close to the value predicted.

Information on square pyramidal Mo_5_ clusters in the literature is limited: until recently, the only known structural data were for (Bu_4_N)_2_[Mo_5_Cl_8_Cl_5_] [28] and (BTMA)_2_[Mo_5_Cl_8_Cl_5_] (BTMA = benzyltrimethylammonium) [29]. These compounds were studied using EPR and had spectra characterized using axially symmetric g-tensors but with g_zz_ > g_xx_ = g_yy_ [23]. In our previous work [24], the magnetic properties of [{Mo_5_(μ_3_-Se)^i^_4_(μ_4_-Se)^i^(μ-pz)^i^_4_}(pzH)^a^_5_]Br·4pzH complex were also analyzed. The polycrystalline EPR spectrum of compound shows g_xx_ = g_yy_ = 2.20 and g_zz_ = 1.99 (g_iso_ = 2.13) at 77K. While the relationship between the *g*_||_ and *g*_⊥_ is determined by the coordination of the metal ions and the nature of the ligands, the deviation of the g-tensor eigenvalues from the free electron g = 2.0023 generally reflects the contribution of orbital motion to the momentum. Thus, the change from selenium to sulfur in the cluster core {Mo_5_(μ_3_-Q)^i^_4_(μ_4_-Q)^i^(μ-pz)^i^_4_}^+^ results in a decrease in g_iso_ from 2.13 for Q = Se to 2.07 for Q = S. The DFT calculations for the [{Mo_5_(μ_3_-S)_4_(μ_4_-S)(μ-pz)_4_}(pzH)_5_]^+^ cation were additionally performed (Appendix A); the obtained values g_xx_ = 2.14, g_yy_ = 2.12 and g_zz_ = 1.98 (g_iso_ = 2.08) agree well with those determined experimentally, supporting the EPR results.

Redox properties of compounds obtained were studied using cyclic voltammetry (CV) in dichloromethane (DCM) or acetonitrile (ACN) (Figure 12 and Appendix A). Main reduction and oxidation potentials are summarized in Table 2. All compounds obtained show two consecutive one-electron reductions corresponding to the reduction of the Mo_5_ cluster from 14 VEC (Mo^III^_4_Mo^IV^) to 15 VEC (Mo^III^_5_) and 16 VEC (Mo^II^Mo^III^_4_). A comparison of the redox properties of the **[1^ox^]Br_2_** cluster with the previously published selenide cluster [{Mo_5_(μ_3_-Se)^i^_4_(μ_4_-Se)^i^(μ-pz)^i^_4_}(pzH)^t^_5_]Br·4pzH (denoted as **Mo_5_Se_5_^red^**) [24], an insignificant effect of the chalcogen in the cluster core {Mo_5_(μ_3_-Q)^i^_4_(μ_4_-Q)^i^(μ-pz)^i^_4_} (Q = S, Se) on the reduction potentials: the first transition is shifted 30 mV to the larger potential, while the second transition remains unchanged. This observation points to the metal-centered nature of the redox transitions, which has also been observed for a number of octahedral chalcogenide clusters of molybdenum and tungsten [27,30,31]. Substitution of the apical pyrazole ligand with bromine shifts the reduction potentials by 250 mV toward lower potentials, while the second reduction becomes irreversible. Bromination of the pyrazolate ligands, on the other hand, shifts the reduction bands by 150 mV toward higher potentials. When both modifications are carried out simultaneously, a superposition of the two shifts is observed, namely a decrease in the reduction potentials by 50 mV compared to **[1^ox^]Br_2_**, while the second transition remain quasi-reversible. Despite the minimal effect of changes in the inner chalcogen ligand on the redox properties, the modification of organic ligands has a greater effect. Bromination of the inner pyrazolate ligands led to a decrease in the ligand-donating ability and, consequently, the complex became easier to reduce, i.e., the reduction potential of the complex was increased. Substitution of the apical pyrazole with bromine apparently increased the electron saturation of the cluster, leading to a decrease in the reduction potential of the cluster.

### 2.4. UV-Vis-NIR Absorption

The absorption properties of the obtained compounds were studied in solution and in solid state in the range 240–2000 nm (Figure 13). In solution, the compounds are characterized by strong absorption in the UV and near visible region (~400 nm). The spectra of the compounds contain two absorption bands in the visible region (400–440 nm and 600–660 nm) and one band in the near-IR region (800–890 nm) (Appendix A), accompanied by less intense absorption up to ~1500 nm. An exception is **[1^red^]Br** (the only 15 VEC cluster), for which there is no pronounced absorption band in the near-IR region, but there is a more intense absorption in almost the entire range studied. The transition from **[1^red^]Br** to **[1^ox^]Br_2_** is accompanied by a bathochromic shift in the first absorption band and a hypsochromic shift in the second (Δ~20 nm). Bromination of pyrazolate ligands (from **[1^ox^]Br_2_** to **[3]Br_2_**) leads to a bathochromic shift of the absorption spectrum (Δ~10 nm) and a strong broadening of the second absorption band. Substitution of the apical pyrazole with bromine (from **[1^ox^]Br_2_** to **[2]Br** and from **[3]Br_2_** to **[4]Br**) leads to a bathochromic shift in the first two absorption bands (Δ~10 nm) and a strong shift in the third band (Δ~80 nm). In this case, the absorption in the region of 1000–1500 nm is practically not changed by the modification of the ligand environment. The absorption spectra in a solid, on the other hand, do not have pronounced absorption bands. Nevertheless, the spectra are characterized by strong absorption in the UV region of the spectrum and less intense absorption up to ~1700 nm for **[1^ox^]Br_2_**, **[2]Br**, **[3]Br_2_**, and **[4]Br** and throughout the spectrum for **[1^red^]Br**. Since, according to quantum chemical calculations (Appendix A, Appendix A), not only metal and chalcogen but also organic ligands contribute to the HOMO and LUMO orbitals of cluster complexes, a change in the ligand environment will affect the energy of the orbitals and their organization and, consequently, affects the absorption spectra. Thus, a simple chemical modification, allows for the absorption of the complexes to be tailored to the required application.

In this work, spectroelectrochemical (SEC) experiments were also carried out to demonstrate the possibility of a rapid change in the absorption spectrum of the complexes by applying electric potential (Figure 14). For all compounds, the transition from 14 VEC (Mo^III^_4_Mo^IV^) to 15 VEC (Mo^III^_5_) was studied, with the exception of **[2]Br**, the oxidation of which leads to the precipitation of a neutral, sparingly soluble form [{Mo_5_(μ_3_-S)^i^_4_(μ_4_-S)^i^(μ-pz)^i^_4_}(pzH)^bs^_4_Br^a^], making it difficult to record the spectra. In all cases, a decrease in absorption in the NIR region and a shift in the absorption band in the visible region are observed during the one-electron reduction. The absorption spectrum obtained in the case of **[1^ox^]Br_2_** is in good agreement with that described above for **[1^red^]Br**, confirming the transition from 14 to 15 VEC cluster. The absorption spectra obtained for the other two complexes should correspond to the reduced forms [{Mo_5_(μ_3_-S)^i^_4_(μ_4_-S)^i^(μ-4-Br-pz)^i^_4_}(pzH)^t^]Br and [{Mo_5_(μ_3_-S)^i^_4_(μ_4_-S)^i^(μ-4-Br-pz)^i^_4_}(pzH)^bs^_4_Br^a^]. Thus, the one-electron reduction makes it possible to control the absorption in the NIR region.

Previously, the absorption of a number of cluster compounds of molybdenum, tantalum and niobium in the near-IR region has been reported [27,32,33,34], in addition to the absorption typical of most cluster complexes in the UV region. The presence of such properties made it possible to consider cluster complexes as components of the UV and NIR barrier. Such barriers can help both to reduce exposure to UV light and/or to save energy through thermal insulation by controlling solar IR radiation [35]. The compounds studied in this work exhibit a significantly broader absorption in the NIR region. Moreover, the absorption of compounds can be changed both by modifying the ligand environment and by reversible redox transitions. In addition to the tuning of optical properties, such clusters exhibit a change in magnetic properties. Such a combination of properties can serve as a basis for the creation of molecular switches based on five-nuclear clusters, which is undoubtedly the goal of our further research.

## 3. Materials and Methods

### 3.1. Chemicals and Materials

Mo_6_Br_12_ was obtained by a reaction of metallic molybdenum with bromine [36]. (Bu_4_N)_2_[Mo_6_I_14_] was synthesized according to the literature procedure [37]. All other reactants and solvents were purchased from Fisher, Alfa Aesar and Sigma-Aldrich and used as received. Drying of Na_2_S: commercial sodium sulfide nonahydrate (>60% Na_2_S) was heated to 200 °C for 90 min using a vacuum pump and held at this temperature for 120 min. The resulting product contains no more than 5% water as determined by TGA.

### 3.2. Syntheses

#### 3.2.1. [{Mo_5_(μ_3_-S)^i^_4_(μ_4_-S)^i^(μ-pz)^i^_4_}(pzH)^t^_5_]Br pzH·H_2_O (Denoted as [1^red^]Br)

Mo_6_Br_12_ (100 mg, 65 μmol), Na_2_S (30 mg, 384 μmol), S_8_ (7 mg, 27 μmol) and pyrazole (200 mg, 2.938 mmol) were grounded and heated in a sealed at ambient conditions glass tube at 200 °C for 2 days. The reaction mixture was slowly cooled to room temperature with the rate of 7.5 °C/h and washed with diethyl ether. Dark-green crystals of **[1^red^]Br** suitable for X-ray structural analyses were separated manually from reaction mixture. Yield: 30 mg (33% based on Mo_6_Br_12_). Anal. Calcd. for C_30_H_38_BrMo_5_N_20_OS_5_: C, 25.5; H, 2.7; N, 19.8; S, 11.3. Found: C, 25.6; H, 2.6; N, 19.4; S, 11.6. EDS: Mo:S:Br atomic ratio was equal to 5:4.9:0.9. FTIR (KBr, cm^−1^): all expected peaks for the pyrazole ligand were observed (Appendix A). The TGA revealed a weight loss of ∼1.6% from 25 to 160 °C (the calculated weight loss of H_2_O is 1.3%) and stability of complex up to 200 °C followed by release of solvate pzH molecule and decomposition of complex (Appendix A). The complex can also be obtained by reaction of Mo_6_Br_12_ (200 mg, 130 μmol), Na_2_S (61 mg, 782 μmol), S_8_ (15 mg, 58 μmol) and pyrazole (178 mg, 2.615 mmol) under same conditions according to PXRD of reaction mixture (Appendix A). However, in this case, separation of the crystals from the reaction mixture is difficult, and upon dissolving of the cluster in organic solvents it oxidizes to **[1^ox^]Br_2_** (see below).

#### 3.2.2. [{Mo_5_(μ_3_-S)^i^_4_(μ_4_-S)^i^(μ-pz)^i^_4_}(pzH)^t^_5_]Br_2_ 2H_2_O (Denoted as **[1^ox^]Br_2_**)

Mo_6_Br_12_ (200 mg, 130 μmol), Na_2_S (61 mg, 782 μmol), S_8_ (15 mg, 58 μmol) and pyrazole (178 mg, 2.615 mmol) were heated in a sealed at ambient conditions glass tube at 200 °C for 2 days. The reaction mixture was slowly cooled to room temperature with the rate of 7.5 °C/h. As noted above, complex **[1^red^]Br** is formed at this stage. The reaction mixture was washed with diethyl ether and dissolved in 200 mL of acetonitrile, which is accompanied by oxidation of the complex with atmospheric oxygen. The solution was filtered off and evaporated until dryness. Powder was dissolved in 100 mL of dichloromethane, filtered off and evaporated until dryness. The desired green product was washed with diethyl ether and dried in air. Yield: 125 mg (66% based on Mo_6_Br_12_). Anal. Calcd. for C_27_H_36_Br_2_Mo_5_N_18_O_2_S_5_: C, 22.4; H, 2.5; N, 17.5; S, 11.1. Found: C, 22.4; H, 2.4; N, 17.5; S, 11.0. EDS: Mo:S:Br atomic ratio was equal to 5:5.2:2.3. FTIR (KBr, cm^−1^): all expected peaks for the pyrazole ligand were observed (Appendix A). The TGA revealed a weight loss of ∼3.1% from 25 to 110 °C (the calculated weight loss of 2 H_2_O is 2.5%) and stability of complex up to 160 °C (Appendix A). ^1^H NMR (500 MHz, CD_3_OD) δ 5.84 (t, 1H, J = 2.35 Hz, H4-pzH^a^), 6.26 (d, 1H, J = 1.98 Hz, H3-pzH^a^), 6.34 (t, 4H, J = 1.90 Hz, H4-pz), 6.90 (t, 4H, J = 2.30 Hz, H4-pzH^bs^), 7.16 (d, 1H, J = 2.27 Hz, H5-pzH^a^), 7.67 (d, 8H, J = 1.98 Hz, H3-, H5-pz), 8.32 (d, 4H, J = 2.27 Hz, H5-pzH^bs^), 8.75 (d, 4H, J = 1.86 Hz, H3-pzH^bs^). ^13^C NMR (126 MHz, CD_3_OD) δ 105.85 (C4-pzH^a^), 106.94 (C4-pzH^bs^), 108.76 (C4-pz), 132.11 (C5-pzH^a^), 132.74 (C5-pzH^bs^), 138.95 (C3-, C5-pz), 145.40 (C3-pzH^a^), 146.50 (C3-pzH^bs^). ^15^N NMR (51 MHz, HCONH_2_) δ 211.4 (NH-pzH^bs^ and NH-pzH^a^), 220.4 (N-pzH^a^), 237.5 (N-pzH^bs^), 257.1 (N-pz). HR-ESI-MS (+) acetonitrile: 1328.6111 ({Mo_5_S_5_(pz)_4_(pzH)_5_Br}^1+^), 1301.5928 ({Mo_5_S_5_(pz)_4_(pzH)_4_(CH_3_CN)Br}^1+^), 1260.5739 ({Mo_5_S_5_(pz)_4_(pzH)_4_Br}^1+^), 1249.6910 ({Mo_5_S_5_(pz)_4_(pzH)_5_}^1+^), 1222.6822 ({Mo_5_S_5_(pz)_4_(pzH)_4_(CH_3_CN)}^1+^), 1181.6564 ({Mo_5_S_5_(pz)_4_(pzH)_4_}^1+^), 624.8470 ({Mo_5_S_5_(pz)_4_(pzH)_5_}^2+^), 590.8262 ({Mo_5_S_5_(pz)_4_(pzH)_4_}^2+^) (Appendix A). To investigate the structure of cationic cluster, the single crystals of [{Mo_5_(μ_3_-S)^i^_4_(μ_4_-S)^i^(μ-pz)^i^_4_}(pzH)^t^_5_][Mo_6_I_14_]·4CH_3_CN (denoted as **[1^ox^][Mo_6_I_14_]**) suitable for X-ray structural analyses were obtained using counter-diffusion in H-form tube of solution of **[1^ox^]Br_2_** in acetonitrile and solution of (Bu_4_N)_2_[Mo_6_I_14_] in acetonitrile.

#### 3.2.3. [{Mo_5_(μ_3_-S)^i^_4_(μ_4_-S)^i^(μ-pz)^i^_4_}(pzH)^bs^_4_Br^a^]Br (Denoted as **[2]Br**)

**[1^ox^]Br_2_** (100 mg, 69 μmol) was dissolved in 5 mL of methanol and 42 μL (348 μmol) of concentrated HBr (46%) was added to the solution. The reaction mixture was heated at 80 °C for one day. The resulting precipitate was separated from the solution, washed with methanol and dichloromethane. The complex was dissolved in 5 mL of DMF, precipitated and washed with diethyl ether and dried in air. Yield: 55 mg (59% based on **[1^ox^]Br_2_**). Anal. Calcd. for C_24_H_28_Br_2_Mo_5_N_16_S_5_: C, 21.5; H, 2.1; N, 16.7; S, 12.0. Found: C, 21.6; H, 2.3; N, 16.9; S, 11.8. EDS: Mo:S:Br atomic ratio was equal to 5:5.0:2.1. FTIR (KBr, cm^−1^): all expected peaks for the pyrazole ligand were observed (Appendix A). ^1^H NMR (500 MHz, DMSO) δ 6.24 (t, 1H, J = 2.02 Hz, H4-pz), 6.96 (q, 1H, J = 2.20 Hz, H4-pzH^bs^), 7.71 (d, 2H, J = 2.04 Hz, H3-, H5-pz), 8.59 (t, 1H, J = 1.75 Hz, H5-pzH^bs^), 8.68 (td, 1H, J_1_ = 2.05 Hz, J_2_ = 0.60 Hz, H3-pzH^bs^), 13.82 (br s, 1H, NH-pzH^bs^). ^13^C NMR (126 MHz, DMSO) δ 106.93 (C4-pzH^bs^), 108.21 (C4-pz), 133.25 (C5-pzH^bs^), 138.81 (C3-, C5-pz), 146.20 (C3-pzH^bs^). ^15^N NMR (51 MHz, HCONH_2_) δ 215.9 (NH-pzH^bs^), 240.5 (N-pzH^bs^), 258.9 (N-pz). HR-ESI-MS (+) methanol: 1260.5692 ({Mo_5_S_5_(pz)_4_(pzH)_4_Br}^1+^), 1260.5692 ({[Mo_5_S_5_(pz)_4_(pzH)_4_Br]_2_}^2+^), 1236.1217 ({[Mo_5_S_5_(pz)_4_(pzH)_4_]_2_Br(CH_3_O)}^2+^), 1212.6760 ({Mo_5_S_5_(pz)_4_(pzH)_4_(CH_3_O)}^1+^), 1192.5229 ({Mo_5_S_5_(pz)_4_(pzH)_3_Br}^1+^), 599.8332 ({Mo_5_S_5_(pz)_4_(pzH)_4_(H_2_O)}^2+^), 590.8263 ({Mo_5_S_5_(pz)_4_(pzH)_4_}^2+^), 556.8154 ({Mo_5_S_5_(pz)_4_(pzH)_3_}^2+^) (Appendix A). The single crystals of [{Mo_5_(μ_3_-S)^i^_4_(μ_4_-S)^i^(μ-pz)^i^_4_}(pzH)^bs^_4_Br^a^]Br·2.5H_2_O·0.5DMF (denoted as **[2]Br**·2.5H_2_O·0.5DMF) suitable for X-ray structural analysis were obtained by vapor diffusion of the diethyl ether into the solution of cluster in mixture of DMF and methanol (90:10 vol. %).

#### 3.2.4. [{Mo_5_(μ_3_-S)^i^_4_(μ_4_-S)^i^(μ-4-Br-pz)^i^_4_}(pzH)^t^_5_]Br_2_ (Denoted as **[3]Br_2_**)

Twenty milliliters of solution of 0.1M Br_2_ in CH_2_Cl_2_ was added to 100 mL solution of **[1^ox^]Br_2_** in CH_2_Cl_2_ (200 mg, 138 μmol). The reaction mixture was sonicated for 10 min. The solution was concentrated to 10 mL and the desired product was precipitated and washed with diethyl ether and dried in air. Yield: 171 mg (72% based on **[1^ox^]Br_2_**). Anal. Calcd. for C_27_H_28_Br_6_Mo_5_N_18_S_5_: C, 18.8; H, 1.6; N, 14.6; S, 9.3. Found: C, 18.6; H, 1.7; N, 14.4; S, 9.2. EDS: Mo:S:Br atomic ratio was equal to 5:5.1:6.1. FTIR (KBr, cm^−1^): all expected peaks for the pyrazole ligand were observed (Appendix A). The TGA revealed a stability of complex up to 140 °C (Appendix A). ^1^H NMR (500 MHz, CD_3_OD) δ 5.86 (t, 1H, J = 2.1 Hz, H4-pzH^a^), 6.21 (br.s, 1H, J = 2.06 Hz, H3-pzH^a^), 6.93 (t, 4H, J = 2.2 Hz, H4-pzH^bs^), 7.18 (d, 1H, J = 2.16 Hz, H5-pzH^a^), 7.89 (s, 8H, H3-, H5-pz), 8.35 (d, 4H, J = 2.30 Hz, H5-pzH^bs^), 8.78 (d, 4H, J = 1.60 Hz, H3-pzH^bs^). ^13^C NMR (126 MHz, CD_3_OD) δ 98.26 (C4-pz), 107.42 (C4-pzH^a^), 108.69 (C4-pzH^bs^), 133.74 (C5-pzH^a^), 134.63 (C5-pzH^bs^), 141.34 (C3-, C5-pz), 148.20 (C3-pzH^bs^). HR-ESI-MS (+) dichloromethane: 1644.2265 ({Mo_5_S_5_(4-Br-pz)_4_(pzH)_5_Br}^1+^), 1565.3101 ({Mo_5_S_5_(4-Br-pz)_4_(pzH)_5_}^1+^), 782.6539 ({Mo_5_S_5_(4-Br-pz)_4_(pzH)_5_}^2+^), 748.1327 ({Mo_5_S_5_(4-Br-pz)_4_(pzH)_4_}^2+^) (Appendix A). The single crystals of [{Mo_5_(μ_3_-S)^i^_4_(μ_4_-S)^i^(μ-Br-pz)^i^_4_}(pzH)^t^]Br_2_·CH_3_CN·Et_2_O (denoted as **[3]Br_2_**·CH_3_CN·Et_2_O) suitable for X-ray structural analysis were obtained by vapor diffusion of the diethyl ether into the solution of cluster in acetonitrile.

#### 3.2.5. [{Mo_5_(μ_3_-S)^i^_4_(μ_4_-S)^i^(μ-4-Br-pz)^i^_4_}(pzH)^bs^_4_Br^a^]Br (Denoted as **[4]Br**)

The same synthesis procedure as for **[2]Br** was performed using **[3]Br_2_** (100 mg, 58 μmol) instead of **[1^ox^]Br_2_** and 35 μL (290 μmol) of concentrated HBr. Yield: 78 mg (81% based on **[3]Br_2_**). Anal. Calcd. for C_24_H_24_Br_6_Mo_5_N_16_S_5_: C, 17.4; H, 1.5; N, 13.5; S, 9.7. Found: C, 17.6; H, 1.6; N, 13.3; S, 9.7. EDS: Mo:S:Br atomic ratio was equal to 5:5.2:6.1. FTIR (KBr, cm^−1^): all expected peaks for the pyrazole ligand were observed (Appendix A). ^1^H NMR (500 MHz, DMSO) 7.01 (br.s, 1H, H4-pzH^bs^), 8.08 (br.s, 2H, H3-, H5-pz), 8.66 (br.s, 1H, H5-pzH^bs^), 8.77 (br.s, 1H, H3-pzH^bs^), 13.91 (br.s, 1H, NH-pzH^bs^). ^13^C NMR (126 MHz, CD_3_OD) δ 94.93 (C4-pz), 107.19 (C4-pzH^bs^), 133.62 (C5-pzH^bs^), 138.81 (C3-, C5-pz), 146.20 (C3-pzH^bs^). ^15^N NMR (51 MHz, HCONH_2_) δ 215.9 (NH-pzH^bs^), 240.5 (N-pzH^bs^), 258.9 (N-pz). HR-ESI-MS (+) methanol: 1576.1998 ({Mo_5_S_5_(4-Br-pz)_4_(pzH)_4_Br}^1+^), 1576.1876 ({[Mo_5_S_5_(4-Br-pz)_4_(pzH)_4_Br]_2_}^2+^), 1551.7522 ({[Mo_5_S_5_(4-Br-pz)_4_(pzH)_4_]_2_(CH_3_O)Br}^2+^), 1528.3044 ({Mo_5_S_5_(4-Br-pz)_4_(pzH)_4_(CH_3_O)}^1+^), 1497.2807 ({Mo_5_S_5_(4-Br-pz)_4_(pzH)_4_}^1+^), 764.6495 ({Mo_5_S_5_(4-Br-pz)_4_(pzH)_4_(CH_3_O)}^2+^), 757.6440 ({Mo_5_S_5_(4-Br-pz)_4_(pzH)_4_(H_2_O)}^2+^), 748.6399 ({Mo_5_S_5_(4-Br-pz)_4_(pzH)_4_}^2+^) (Appendix A). The single crystals of [{Mo_5_(μ_3_-S)^i^_4_(μ_4_-S)^i^(μ-4-Br-pz)^i^_4_}(pzH)^bs^_4_Br^a^]Br·1.25H_2_O·2DMSO (denoted as **[4]Br**·1.25H_2_O·2DMSO) suitable for X-ray structural analysis were obtained by vapor diffusion of the ethyl acetate into the solution of cluster in DMSO.

### 3.3. Physical Methods

Elemental analyses were obtained using a EuroVector EA3000 Elemental Analyser. FTIR spectra were recorded on a Bruker Vertex 80 as KBr disks. Energy-dispersive X-ray spectroscopy (EDS) was performed on a Hitachi TM3000 TableTop SEM with Bruker QUANTAX 70 EDS equipment. The thermal properties (TGA) were studied on a Thermo Microbalance TG 209 F1 Iris (NETZSCH) from 25 to 850 °C at the heating rate of 10 °C·min^−1^ in He flow (30 mL·min^−1^). Powder X-ray diffraction (PXRD) patterns were collected on a Philips PW 1820/1710 diffractometer (CuK_α_ radiation, graphite monochromator and Si as an external reference). UV-vis-NIR absorption spectra in solution and diffuse reflectance spectra in solid state were recorded using a UV-Vis-NIR 3101 PC spectrophotometer (Shimadzu Corporation, Kyoto, Japan). Absorption spectra for spectroelectrochemical measurements were recorded on a Cary 60 UV-Vis Spectrophotometer (Agilent).

The high-resolution electrospray mass spectrometric (HR-ESI-MS) detection was performed at the Center of Collective Use «Mass spectrometric investigations» SB RAS in positive mode within 500–3000 m/z range on an electrospray ionization quadrupole time-of-flight (ESI-Q-TOF) high-resolution mass spectrometer Maxis 4G (Bruker Daltonics, Germany). The 1D and 2D NMR spectra of sample were obtained from CD_3_OD or DMSO-d_6_ solutions at room temperature on a Bruker Avance III 500 FT-spectrometer with working frequencies 500.03, 125.73 and 50.67 MHz for ^1^H, ^13^C and ^15^N, respectively. The ^1^H and ^13^C NMR chemical shifts are reported in ppm of the δ scale and referred to the signal of the methyl group of the solvent (δ = 3.31 ppm for residual protons for the ^1^H- and 49.0 ppm for ^13^C-NMR spectra for CD_3_OD or δ = 2.50 ppm for residual protons for the ^1^H- and 39.50 ppm for ^13^C-NMR spectra for DMSO-d_6_). The ^15^N NMR chemical shifts are referred to external standard formamide (δ (^15^N) = 112.5 ppm). ^15^N NMR spectrum was obtained as projection of 2-D ^1^H–^15^N correlation. Assignment of the signals were carried out using 2D (HSQC, HMBC) NMR techniques.

### 3.4. Single-Crystal X-ray Diffraction Analysis (XRD)

Single-crystal X-ray diffraction data for **[1^red^]Br** were collected at 150 K on a Bruker Nonius X8Apex CCD area-detector Bruker Apex DUO diffractometer and for **[1^ox^][Mo_6_I_14_]**, **[2]Br**·2.5H_2_O·0.5DMF, **[3]Br_2_**·CH_3_CN·Et_2_O and **[4]Br**·1.25H_2_O·2DMSO at 150K on a Bruker D8 VENTURE fitted with graphite-monochromatized MoKα radiation (λ = 0.71073 Å). Absorption corrections were made empirically using the SADABS program [38]. The structures were solved using the direct method and further using by the full-matrix least-squares method using the SHELXTL program package [38]. All nonhydrogen atoms were refined anisotropically. Appendix A summarizes crystallographic data, while CCDC 2277466-2277470 contain the supplementary crystallographic data for this paper. These data can be obtained free of charge from the Cambridge Crystallographic Data Centre via www.ccdc.cam.ac.uk/data_request/cif (accessed on 28 June 2023).

### 3.5. Cyclic Voltammetry

Cyclic voltammetry was carried out with Elins P-20X8 voltammetry analyzer using three-electrode scheme with GC working, Pt auxiliary and Ag/AgCl/3.5M KCl reference electrodes. Investigations were carried out for 5·10^−4^ M solution of corresponding cluster compound in 0.1 M Bu_4_NClO_4_ in acetonitrile or dichloromethane under Ar atmosphere.

### 3.6. EPR

The X-band continuous-wave EPR spectra were recorded at 77 and 300 K with a Varian E-109 spectrometer. The frequency of the spectrometer was calibrated with a 2,2-diphenyl-1-picrylhydrazyl (DPPH) standard sample. The weighted portion of copper(II) sulfate pentahydrate (CuSO_4_·5H_2_O) was used to evaluate the concentration of paramagnetic species. The EPR spectra were simulated in the MATLAB program package with the EasySpin toolbox [39].

### 3.7. Magnetic Susceptibility

The magnetic susceptibility of the sample studied was measured using the Faraday technique in the temperature range of 80–300 K. The measurement procedure included two cycles: cooling from 300 to 80 K and heating from 80 to 300 K. Then, the obtained values were averaged. The temperature was stabilized using a Delta DTB9696 temperature controller. The signal from the magnetometer was measured using a high-precision digital Keysight 34465A voltmeter. The magnetic field strength was of 8.2 kOe. The powder sample weighing of 66.8 mg in an open quartz ampoule was placed in the measurement cell of the magnetometer. In order to remove paramagnetic oxygen molecules, the measurement cell was vacuumed to 0.01 Torr pressure. Then, the measurement cell was filled with a helium with a pressure of 5 Torr.

### 3.8. DFT Calculations

The g-tensor has been calculated as a second derivative property (spin-orbit coupling and external magnetic field as perturbation) [40] within the two-component relativistic zeroth-order regular approximation (ZORA) [41] implemented in the ADF2022 software package (version 2022.105) [42,43]. This calculation was performed with hybrid B3LYP functional [44] and all-electron basis sets TZP for all atoms [45]. In this case, the structure of [{Mo_5_(μ_3_-S)_4_(μ_4_-S)(μ-pz)_4_}(pzH)_5_]^+^ was taken from structural data and was not optimized afterward.

## 4. Conclusions

In summary, a one-step synthesis of square pyramidal molybdenum chalcogenide clusters [{Mo_5_(μ_3_-S)_4_(μ_4_-S)(μ-pz)_4_}(pzH)_5_]^1+/2+^ from an octahedral cluster Mo_6_Br_12_ is demonstrated. The cationic 1+ charged cluster obtained during ampoule synthesis at 200 °C contains 15 VEC and is a paramagnetic complex with a calculated effective moment value of μ_eff_ = 2.19(3) μ_B_ and g_iso_ = 2.07 (g_xx_ = 2.12, g_yy_ = 2.11, and g_zz_ = 1.99) corresponding to one unpaired electron. Upon dissolution, the complex is oxidized to 14 VEC doubly charged form, allowing the compound to be characterized in solution, for example, using NMR spectroscopy. The possibility of modifying the ligand environment of [{Mo_5_(μ_3_-S)_4_(μ_4_-S)(μ-pz)_4_}(pzH)_5_]^2+^ was demonstrated using two approaches: (i) interaction with HBr in methanol leads to the replacement of only the apical pyrazole ligand; (ii) interaction with Br_2_ in methylene chloride leads to the selective bromination of the pyrazolate ligands at the fourth position. The possibility of successive modification of the cluster using both approaches was also demonstrated. Clusters [{Mo_5_(μ_3_-S)^i^_4_(μ_4_-S)^i^(μ-pz)^i^_4_}(pzH)^bs^_4_Br^a^]^+^, [{Mo_5_(μ_3_-S)^i^_4_(μ_4_-S)^i^(μ-4-Br-pz)^i^_4_}(pzH)^t^_5_]^2+^ and [{Mo_5_(μ_3_-S)^i^_4_(μ_4_-S)^i^(μ-4-Br-pz)^i^_4_}(pzH)^bs^_4_Br^a^]^+^ were characterized in detail using a number of physicochemical methods. It has been shown that a change in the ligand environment affects the redox properties of the cluster: the reduction potentials of the cluster core shift to more negative values upon substitution of the apical pyrazole (~250 mV) and vice versa upon bromination of the pyrazolate ligands (~150 mV). In addition, modification of the ligand environment significantly alters the absorption of cluster complexes in the visible and near-IR. Tuning the absorption of the complexes in the visible and near-IR by varying the ligand environment and by reversible redox transitions makes it possible to consider pentanuclear cluster complexes as components of UV and NIR barriers. On the other hand, reversible redox transitions between the diamagnetic and paramagnetic states may allow for the development of new sensors for various external influences. The search for practical applications and a detailed study of the properties of such molybdenum chalcogenide cluster complexes is the goal of our further research.

## Figures and Tables

**Figure 1 ijms-24-13879-f001:**
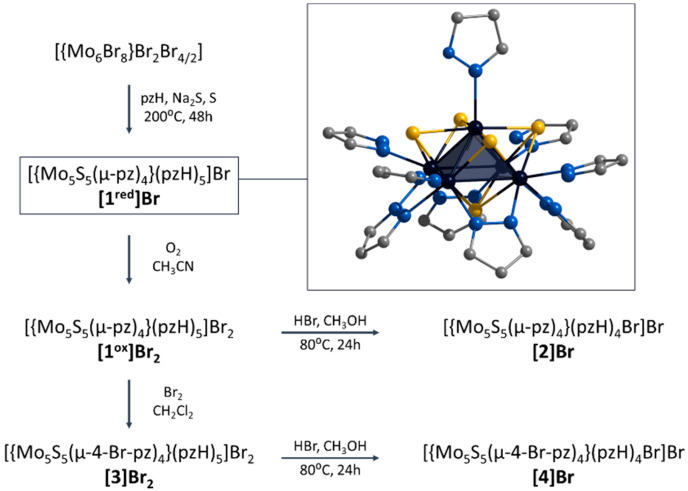
Schematic representation of the compounds obtained in this work and their transformations. Typical structure of [{Mo_5_(μ_3_-S)^i^_4_(μ_4_-S)^i^(μ-pz)^i^_4_}(pzH)^t^_5_]^+^ is demonstrated in frame. Color code: Mo_5_—dark blue square pyramid; S—yellow; N—blue; C—grey. Hydrogen atoms are omitted for clarity.

**Figure 2 ijms-24-13879-f002:**
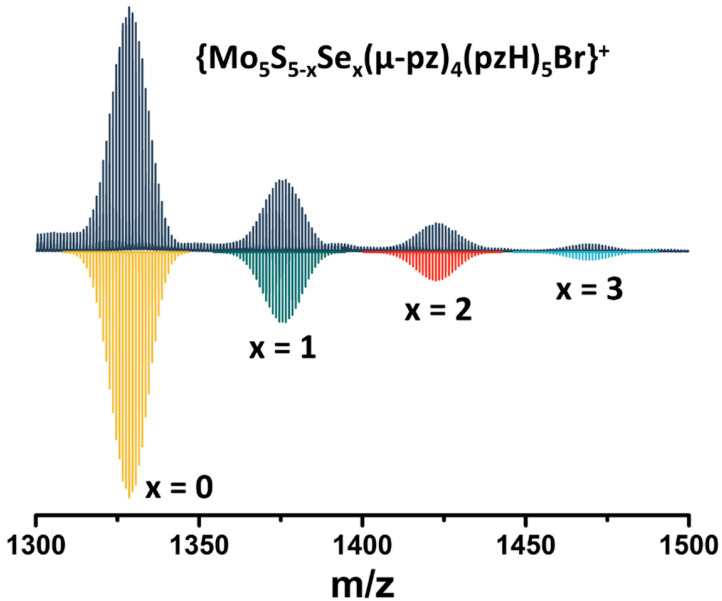
Fragment of mass spectrum of [{Mo_5_S_5−x_Se_x_(μ-pz)_4_}(pzH)_5_]Br_2_ (x = 0, 1, 2, 3) solution in acetonitrile (dark blue) and calculated forms (colored).

**Figure 3 ijms-24-13879-f003:**
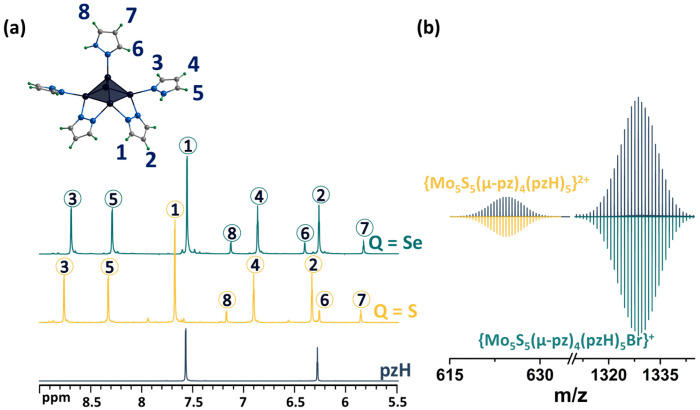
(**a**) ^1^H NMR spectra of pentanuclear clusters [{Mo_5_Q_5_(μ-pz)_4_}(pzH)_5_]^2+^ (Q = S, Se) in CD_3_OD in comparison with pyrazole. (**b**) Fragment of mass spectrum of **[1^ox^]Br_2_** solution in acetonitrile (dark blue) and calculated forms (colored). Numbers in circles on the NMR spectra correspond to the numbers of hydrogen atoms in the structure.

**Figure 4 ijms-24-13879-f004:**
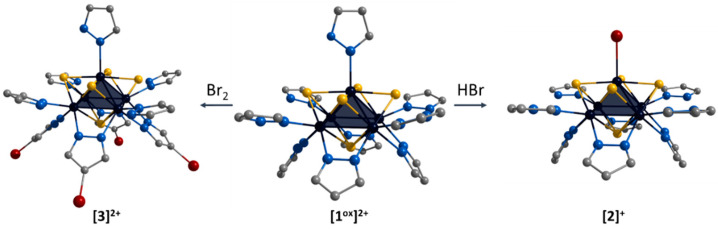
Schematic presentation of ligand environment modification of [{Mo_5_(μ_3_-S)^i^_4_(μ_4_-S)^i^(μ-pz)^i^_4_}(pzH)^t^_5_]^2+^. Color code: Mo_5_—dark blue square pyramid; S—yellow; N—blue; C—grey; Br—red. Hydrogen atoms are omitted for clarity.

**Figure 5 ijms-24-13879-f005:**
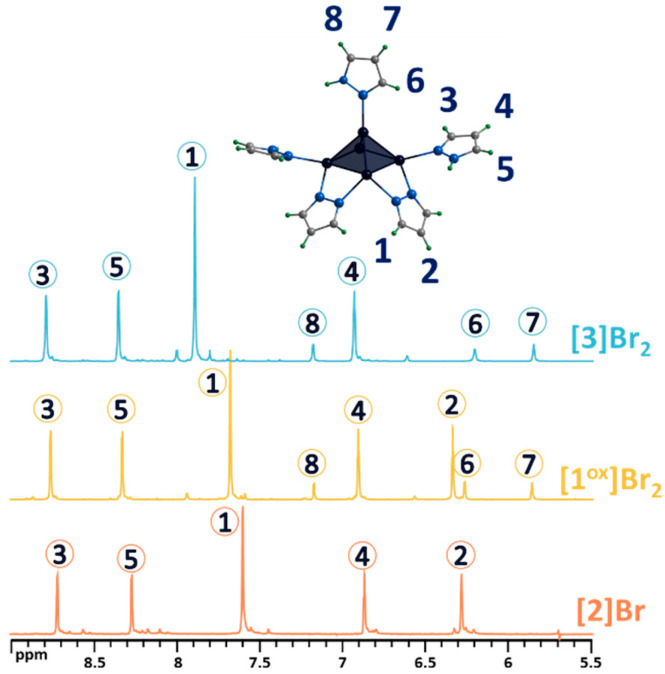
^1^H NMR spectra of pentanuclear clusters **[1^ox^]Br_2_**, **[3]Br_2_** and **[2]Br** in CD_3_OD. Due to the low solubility of **[2]Br** in methanol, the spectrum contains low intensity signals from impurities with higher solubility. Numbers in circles on the NMR spectra correspond to the numbers of hydrogen atoms in the structure.

**Figure 6 ijms-24-13879-f006:**
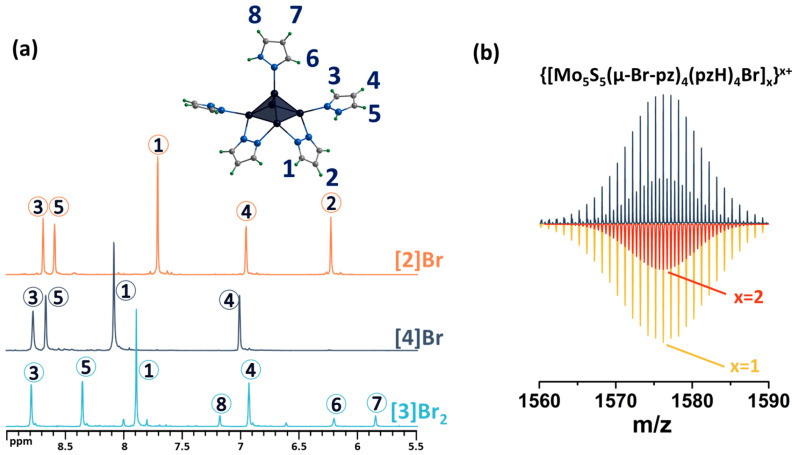
(**a**) ^1^H NMR spectra of **[4]Br** in DMSO-d_6_ in comparison with **[3]Br_2_** in CD_3_OD and **[2]Br** in DMSO-d_6_. (**b**) Fragment of mass spectrum of **[4]Br** solution in acetonitrile (dark blue) and calculated forms (colored). Numbers in circles on the NMR spectra correspond to the numbers of hydrogen atoms in the structure.

**Figure 7 ijms-24-13879-f007:**
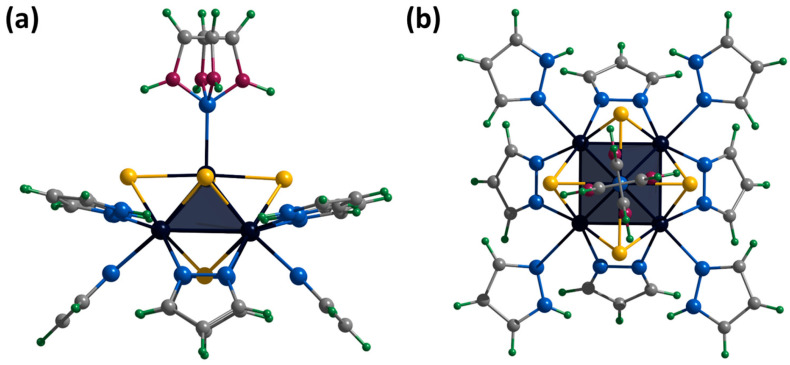
Crystal structure of cationic cluster [{Mo_5_(μ_3_-S)^i^_4_(μ_4_-S)^i^(μ-pz)^i^_4_}(pzH)^t^_5_]^+^: side (**a**) and top (**b**) view. Color code: Mo_5_—dark blue square pyramid; S—yellow; N—blue; C—grey; H—green; disordered C/N—purple.

**Figure 8 ijms-24-13879-f008:**
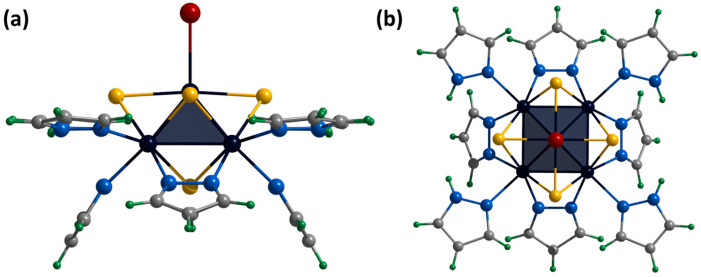
Crystal structure of cationic cluster [{Mo_5_(μ_3_-S)^i^_4_(μ_4_-S)^i^(μ-pz)^i^_4_}(pzH)^bs^_4_Br^a^]^+^: side (**a**) and top (**b**) view. Color code: Mo_5_—dark blue square pyramid; S—yellow; N—blue; C—grey; H—green; Br—red.

**Figure 9 ijms-24-13879-f009:**
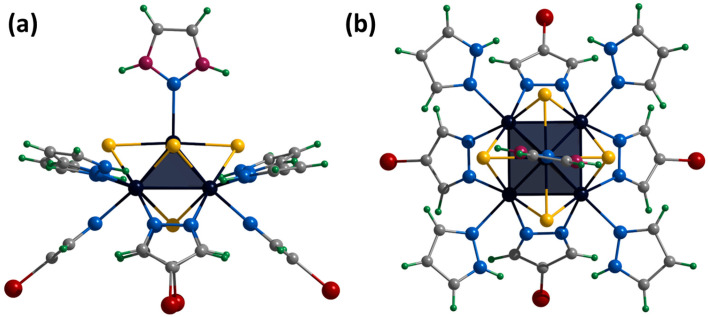
Crystal structure of cationic cluster [{Mo_5_(μ_3_-S)^i^_4_(μ_4_-S)^i^(μ-4-Br-pz)^i^_4_}(pzH)^t^]^2+^: side (**a**) and top (**b**) view. Color code: Mo_5_—dark blue square pyramid, S—yellow, N—blue, C—grey, H—green, Br—red, disordered C/N—purple.

**Figure 10 ijms-24-13879-f010:**
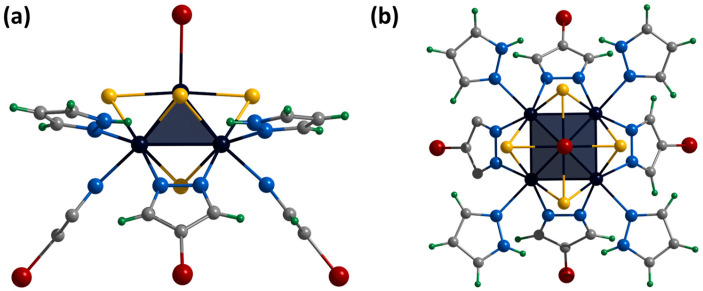
Crystal structure of cationic cluster [{Mo_5_(μ_3_-S)^i^_4_(μ_4_-S)^i^(μ-4-Br-pz)^i^_4_}(pzH)^bs^_4_Br^a^]^+^: side (**a**) and top (**b**) view. Color code: Mo_5_—dark blue square pyramid; S—yellow; N—blue; C—grey; H—green; Br—red.

**Figure 11 ijms-24-13879-f011:**
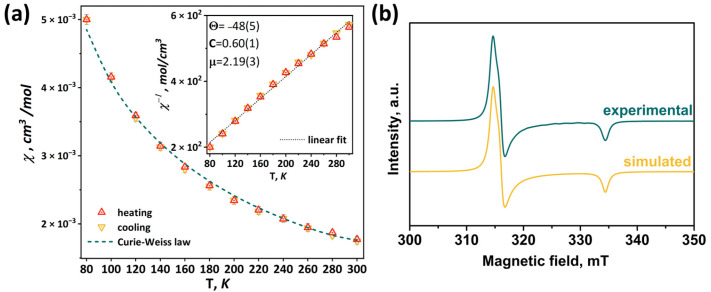
Temperature dependence of magnetic susceptibility (**a**) and EPR spectrum measured at 77 K in comparison with simulated one (**b**) of **[1^red^]Br**. Inset: linear fitting of inverse magnetic susceptibility.

**Figure 12 ijms-24-13879-f012:**
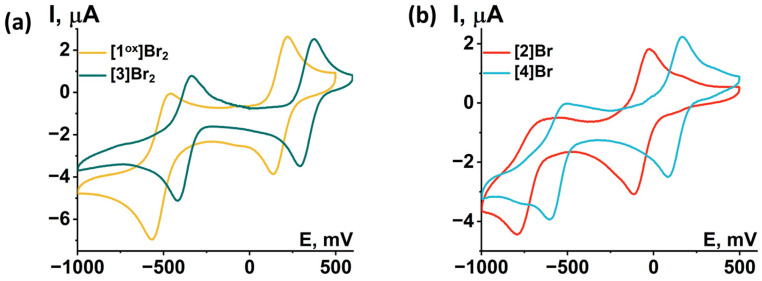
Cyclic voltammetry of the clusters (0.5 mM) in 0.1 M Bu_4_NClO_4_ dichloromethane solution for (**a**) **[1^ox^]Br_2_** and **[3]Br_2_** and (**b**) **[2]Br** and **[4]Br**. Scan rate—100 mV/s. Reference electrode: Ag/AgCl/3.5 M KCl.

**Figure 13 ijms-24-13879-f013:**
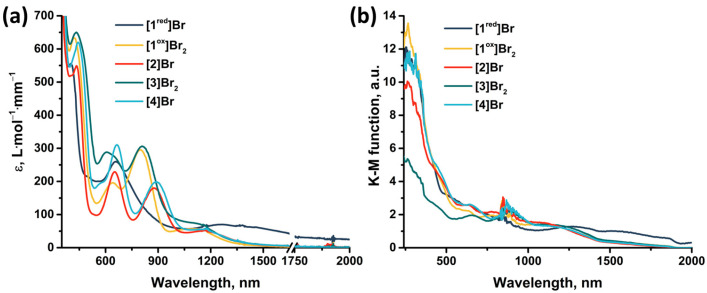
(**a**) UV-vis-NIR spectra of compounds obtained in acetonitrile or DMSO (only for **[4]Br**) solutions. The absorption region 1650–1750 nm is not presented due to the strong absorption of the solvents. (**b**) Diffuse reflectance spectra converted to absorption spectra using the Kubelka–Munk function.

**Figure 14 ijms-24-13879-f014:**
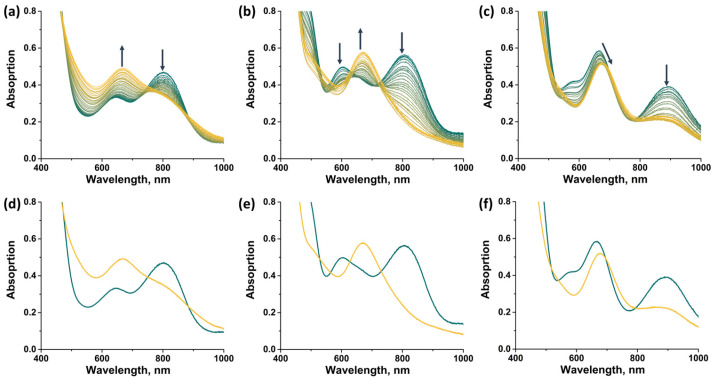
UV-vis-NIR spectra of compounds obtained in dichloromethane solutions by applying an electric potential over time: (**a**) **[1^ox^]Br_2_**, (**b**) **[3]Br_2_**, (**c**) **[4]Br**. Comparison of the first (green) and last (yellow) spectra for **[1^ox^]Br_2_** (**d**), **[3]Br_2_** (**e**) and **[4]Br** (**f**). Arrows indicate direction of change in spectra.

**Table 1 ijms-24-13879-t001:** Selected interatomic and average distances (Å) for compounds obtained in comparison with data from the literature.

Compound	Mo^bs^–Mo^bs^ (Average)	Mo^bs^–Mo^a^ (Average)	Mo–μ_3_-S (Average)	Mo–μ_4_-S (Average)
**[1^red^]Br**	2.7980(3)–2.8010(4) (2.8000)	2.6384(4)–2.6459(4) (2.6422)	2.3982(8)–2.4395(8) (2.4187)	2.4235(6)–2.4318(6) (2.4277)
**[1^ox^][Mo_6_I_14_]**	2.8136(6)–2.8249(6) (2.8193)	2.6288(6)–2.6322(6) (2.6305)	2.3781(14)–2.4523(12) (2.4168)	2.4296(12)–2.4390(12) (2.4343)
**[2]Br·2.5H_2_O·0.5DMF**	2.7951(17)–2.8217(17) (2.8080)	2.6285(18)–2.6345(18) (2.6310)	2.368(4)–2.479(4) (2.423)	2.443(4)–2.447(4) (2.446)
**[3]Br_2_·CH_3_CN·Et_2_O**	2.8073(10)–2.8244(10) (2.8168)	2.6185(10)–2.6341(10) (2.6269)	2.363(2)–2.465(2) (2.402)	2.436(2)–2.449(2) (2.441)
**[4]Br·1.25H_2_O·2DMSO**	2.7816(9)–2.7985(17) (2.790)	2.6468(11)–2.6526(10) (2.6497)	2.387(2)–2.441(3) (2.411)	2.432(2)–2.436(2) (2.434)
**Mo_5_Se_5_^red^**	2.8398(2)	2.6823(3)	2.5572(2)	2.5329(3)
**Mo_5_Se_5_^ox^**	2.865(1)	2.660(1)–2.674(1) (2.667)	2.499(1)–2.5712(8) (2.5326)	2.547(1)–2.550(1) (2.548)
	**Mo–N^μ-pz^** **(average)**	**Mo^bs^–N^pzH^** **(average)**	**Mo^a^–N^pzH^** **(average)**	**Mo^a^–Br** **(average)**
**[1^red^]Br**	2.183(3)–2.194(2) (2.186)	2.207(3)–2.210(3) (2.209)	2.199(4)	–
**[1^ox^][Mo_6_I_14_]**	2.155(4)–2.177(4) (2.165)	2.189(4)–2.198(4) (2.194)	2.197(6)	–
**[2]Br·2.5H_2_O·0.5DMF**	2.145(14)–2.180(15) (2.167)	2.212(12)–2.225(13) (2.216)	–	2.632(3)–2.640(3) (2.636)
**[3]Br_2_·CH_3_CN·Et_2_O**	2.149(7)–2.188(7) (2.175)	2.190(7)–2.209(7) (2.200)	2.236(8)	–
**[4]Br·1.25H_2_O·2DMSO**	2.223(7)	2.189(6)–2.193(6) (2.191)	–	2.6327(15)
**Mo_5_Se_5_^red (a)^**	2.184(2)–2.194(2) (2.189)	2.217(2)	2.239(4)	–
**Mo_5_Se_5_^ox (a)^**	2.164(8)–2.209(7) (2.179)	2.214(7)–2.230(7) (2.222)	2.25(1)	–

^(a)^ data for **Mo_5_Se_5_^red^** = [{Mo_5_(μ_3_-Se)^i^_4_(μ_4_-Se)^i^(μ-pz)^i^_4_}(pzH)^t^_5_]Br·4pzH and **Mo_5_Se_5_^ox^** = [{Mo_5_(μ_3_-Se)^i^_4_(μ_4_-Se)^i^(μ-pz)^i^_4_}(pzH)^t^_5_]Br_2_·2H_2_O are from [24].

**Table 2 ijms-24-13879-t002:** Main electrochemical potentials (Volt vs. Ag/AgCl) for chalcogenide clusters in 0.1 M Bu_4_NClO_4_ acetonitrile (ACN) or dichloromethane (DCM) solutions.

Compound	Solvent	Process [a]	E_a_	E_c_	E_1/2_	Ref.
**Mo_5_Se_5_^red^**	ACN	15 VEC to 16 VEC, qrev	–0.54	–0.60	–0.57	[24]
		15 VEC to 14 VEC, rev	0.14	0.07	0.11	
**[1^ox^]Br_2_**	ACN	14 VEC to 15 VEC, rev	0.17	0.10	0.14	This work
		15 VEC to 16 VEC, qrev	–0.53	–0.61	–0.57	
	DCM	14 VEC to 15 VEC, rev	0.22	0.14	0.18	
		15 VEC to 16 VEC, qrev	–0.46	–0.57	–0.52	
**[2]Br**	DCM	14 VEC to 15 VEC, rev	–0.03	–0.12	–0.08	
		15 VEC to 16 VEC, irrev	–	–0.80	–	
**[3]Br_2_**	DCM	14 VEC to 15 VEC, rev	0.38	0.29	0.33	
		15 VEC to 16 VEC, qrev	–0.34	–0.42	–0.38	
**[4]Br**	DCM	14 VEC to 15 VEC, rev	0.17	0.09	0.13	
		15 VEC to 16 VEC, qrev	–0.51	–0.60	–0.56	

[a] qrev = quasi-reversible; rev = reversible; irrev = irreversible.

## Data Availability

Crystal structure data can be obtained free of charge from The Cambridge Crystallographic Data Centre via www.ccdc.cam.ac.uk/data_request/cif (accessed on 28 June 2023) or are available on request from the corresponding author.

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
