# Peer review of "Chemical Diversity of Mo5S5 Clusters with Pyrazole: Synthesis, Redox and UV-vis-NIR Absorption Properties"

_ijms, 2023, doi:10.3390/ijms241813879_

Round 1
Reviewer 1 Report
The authours present the synthesis, characterization, and UV-vis-NIR absorption properties of Mo5S5 clusters. The characterizations are adequate. The presentation is clear. I suggest the authours to address two issues as given below.
1. Some TDDFT calculations are needed to assign the absorption bands of typical complexes.
2. Since these complexes display well-definced redox processes, please perform spectroelectrochemical measurements to check the spectral changes.
The Quality of English Language is acceptable.
Author Response
The authours present the synthesis, characterization, and UV-vis-NIR absorption properties of Mo5S5 clusters. The characterizations are adequate. The presentation is clear. I suggest the authours to address two issues as given below.
- Some TDDFT calculations are needed to assign the absorption bands of typical complexes.
Answer: Thank you for the helpful suggestion. Undoubtedly, the calculations will help to make a more detailed analysis of the absorption spectra. However, for such objects (cluster compounds with a large number of elements), such experiments are highly time-consuming (about week for one cluster) and cannot be performed within the scope of this review. We will take this fact into account in our future works.
- Since these complexes display well-definced redox processes, please perform spectroelectrochemical measurements to check the spectral changes.
Answer: Thanks for the important suggestion. The spectroelectrochemical measurements were carried out for all cluster obtained and additional discussion was added in the section 2.4. UV-Vis-NIR absorption.
Reviewer 2 Report
The manuscript "Chemical diversity of Mo5S5 clusters with pyrazole: synthesis, redox and UV-vis-NIR absorption properties
" presents a detailed study on the reactivity and ligand environment variation of new square-pyramidal molybdenum chalcogenide clusters. The synthesis of these clusters from an octahedral Mo6Br12 cluster is demonstrated, and the compounds are characterized using various physicochemical methods. The redox properties and absorption in the UV-visible and near-infrared region of the compounds are also studied. The results show that modifying the ligand environment affects the redox properties and absorption of the cluster complexes, making them promising components for UV and NIR barriers. Overall, the manuscript provides valuable insights into the reactivity and properties of square-pyramidal molybdenum chalcogenide clusters. Here are some suggestions to improve the clarity and impact of the manuscript:
1. Introduction: Provide a brief background on the importance and applications of transition metal clusters with promising properties.
2. Methodology: Provide more details on the synthesis procedure for the one-step synthesis of the square-pyramidal molybdenum chalcogenide clusters from the octahedral Mo 6 Br 12 cluster. Include information about the reaction conditions, reagents used, and purification steps.
3. Results and Discussion: Discuss the redox properties and absorption characteristics of the obtained compounds in the UV-visible and near-infrared region. Explain the significance of these properties and how they can be practically useful. Clearly explain the observed shifts in the reduction potentials of the cluster core upon substitution of the apical pyrazole ligand or bromination of the pyrazolate ligands. Provide a thorough analysis of the impact of ligand modification on the redox properties.
4. Conclusion: Discuss the broader implications and potential future directions for research in the field of square-pyramidal molybdenum chalcogenide clusters.
5. General suggestions: Define all abbreviations before their first use and provide explanations for any specialized terminology that may be unfamiliar to a broad readership. Proofread the manuscript carefully to correct any grammatical errors or inconsistencies.
By addressing these suggestions, the manuscript will become more accessible and engaging to readers, while effectively conveying the significance of the research findings on the reactivity and ligand environment of square-pyramidal molybdenum chalcogenide clusters.
In conclusion, the manuscript presents a thorough investigation of the reactivity and ligand environment variation of square-pyramidal molybdenum chalcogenide clusters. The synthesis and characterization of the compounds are well-described, and the results are presented clearly. The discussion of the redox properties and absorption of the clusters adds valuable insights into their potential applications. With some additional background information and a more detailed discussion on the applications of the clusters, the manuscript would be even stronger. Overall, this manuscript makes a significant contribution to the field of transition metal chemistry and would be of interest to researchers in the field.
Author Response
The manuscript "Chemical diversity of Mo5S5 clusters with pyrazole: synthesis, redox and UV-vis-NIR absorption properties" presents a detailed study on the reactivity and ligand environment variation of new square-pyramidal molybdenum chalcogenide clusters. The synthesis of these clusters from an octahedral Mo6Br12 cluster is demonstrated, and the compounds are characterized using various physicochemical methods. The redox properties and absorption in the UV-visible and near-infrared region of the compounds are also studied. The results show that modifying the ligand environment affects the redox properties and absorption of the cluster complexes, making them promising components for UV and NIR barriers. Overall, the manuscript provides valuable insights into the reactivity and properties of square-pyramidal molybdenum chalcogenide clusters. Here are some suggestions to improve the clarity and impact of the manuscript:
- Introduction: Provide a brief background on the importance and applications of transition metal clusters with promising properties.
Answer: The introduction part was extended according reviewer suggestion.
- Methodology: Provide more details on the synthesis procedure for the one-step synthesis of the square-pyramidal molybdenum chalcogenide clusters from the octahedral Mo6Br12 cluster. Include information about the reaction conditions, reagents used, and purification steps.
Answer: The required information was added in Result and Discussion part.
- Results and Discussion: Discuss the redox properties and absorption characteristics of the obtained compounds in the UV-visible and near-infrared region. Explain the significance of these properties and how they can be practically useful. Clearly explain the observed shifts in the reduction potentials of the cluster core upon substitution of the apical pyrazole ligand or bromination of the pyrazolate ligands. Provide a thorough analysis of the impact of ligand modification on the redox properties.
Answer: The discussion and explanation of the results were added in Result and Discussion part.
- Conclusion: Discuss the broader implications and potential future directions for research in the field of square-pyramidal molybdenum chalcogenide clusters.
Answer: The conclusion was extended according reviewer suggestions.
- General suggestions: Define all abbreviations before their first use and provide explanations for any specialized terminology that may be unfamiliar to a broad readership. Proofread the manuscript carefully to correct any grammatical errors or inconsistencies.
Answer: The required corrections were done.
By addressing these suggestions, the manuscript will become more accessible and engaging to readers, while effectively conveying the significance of the research findings on the reactivity and ligand environment of square-pyramidal molybdenum chalcogenide clusters.
In conclusion, the manuscript presents a thorough investigation of the reactivity and ligand environment variation of square-pyramidal molybdenum chalcogenide clusters. The synthesis and characterization of the compounds are well-described, and the results are presented clearly. The discussion of the redox properties and absorption of the clusters adds valuable insights into their potential applications. With some additional background information and a more detailed discussion on the applications of the clusters, the manuscript would be even stronger. Overall, this manuscript makes a significant contribution to the field of transition metal chemistry and would be of interest to researchers in the field.
Answer: Thank you for your positive feedback and relevant suggestions.
Round 2
Reviewer 1 Report
the revised manuscript is acceptable.